# Identification and Functional Characterization of Oxidosqualene Cyclases from Medicinal Plant *Hoodia gordonii*

**DOI:** 10.3390/plants13020231

**Published:** 2024-01-14

**Authors:** Iffat Parveen, Mei Wang, Joseph Lee, Jianping Zhao, Yingjie Zhu, Amar G. Chittiboyina, Ikhlas A. Khan, Zhiqiang Pan

**Affiliations:** 1National Center for Natural Products Research, School of Pharmacy, University of Mississippi, University, MS 38677, USA; 2Natural Products Utilization Research Unit, Agricultural Research Service, United States Department of Agriculture, University, MS 38677, USA; 3Department of Pathology and Laboratory Medicine, Memorial Sloan Kettering Cancer Center, New York, NY 10065, USA; 4Division Pharmacognosy, Department of BioMolecular Sciences, School of Pharmacy, University of Mississippi, University, MS 38677, USA

**Keywords:** lupeol synthase, cycloartenol synthase, *Hoodia gordonii*, lupeol, cycloartenol

## Abstract

Oxidosqualene cyclases (OSCs) are the key enzymes accountable for the cyclization of 2,3-oxidosqualene to varied triterpenoids and phytosterols. *Hoodia gordonii* (from the family Apocynaceae), a native of the Kalahari deserts of South Africa, Namibia, and Botswana, is being sold as a prevalent herbal supplement for weight loss. The appetite suppressant properties are attributed to P57AS3, an oxypregnane steroidal glycoside. At the molecular level, the enzymes involved in the biosynthesis of triterpenes and phytosterols from *H. gordonii* have not been previously reported. In the current study, predicted transcripts potentially encoding oxidosqualene cyclases were recognized first by searching publicly available *H. gordonii* RNA-seq datasets. Two OSC-like sequences were selected for functional analysis. A monofunctional OSC, designated *HgOSC1* which encodes lupeol synthase, and *HgOSC2*, a multifunctional cycloartenol synthase forming cycloartenol and other products, were observed through recombinant enzyme studies. These studies revealed that distinct OSCs exist for triterpene formation in *H. gordonii* and provided opportunities for the metabolic engineering of specific precursors in producing phytosterols in this plant species.

## 1. Introduction

Sterols and triterpenoids comprise a vast set of natural products that are abundant in plants and play important roles in various biological functions. In eukaryotes, sterols are the constituents of cell membranes and serve as the precursors for hormones, thus constituting primary metabolites. Different types of skeletal triterpenes exist in plants and are regarded as secondary metabolites because they are species-specific, and their function is unknown in plants. However, some triterpenes, such as betulinic acid, glycyrrhizin, and avenacins, have medicinal properties and hence these are widely used to mitigate various disorders in traditional medicine [1,2]. The common precursor for both sterols and triterpenes is 2,3-oxidosqualene. The cyclization of oxidosqualene into tetracyclic and pentacyclic carbon skeletons of sterols and triterpenes is catalyzed by oxidosqualene cyclases (OSCs). OSCs are further classified into monofunctional and multifunctional based on single or multi-cyclic products formed after cyclization. 2,3-oxidosqualene cyclization involves carbocationic cyclization followed by a cascade of rearrangement steps that are finally stabilized by deprotonation [3,4,5]. Besides these cyclization events, how 2,3-oxidosqualene is folded into the enzyme active site also determines the diversity of end products. The two conformations chair–boat–chair (CBC) and chair–chair–chair (CCC) generate protosteryl and dammarenyl cations, respectively. The dammarenyl cation is converted to different triterpenes such as α-amyrins, β-amyrins, and lupeol, whereas protosteryl cation leads to the formation of cycloartenol which is the precursor of phytosterols and steroid hormones [4]. Apart from OSCs, the chemical diversity of steroids and triterpenes is also attributed to the catalytic activity of different enzyme families, such as cytochromes P450 (CYPs) oxidoreductases and UDP-dependent glucosyltransferases [6]. However, the OSCs at the branch points are responsible for moving the metabolic profile of the plant and, hence, for the production of different secondary metabolites. Therefore, these OSCs could be used to produce specific secondary metabolites through genetic manipulations. However, detailed characteristics of the candidate OSCs covering different branch points and their mechanism of action are required for the desired genetic manipulation. Several OSCs cDNA involved in sterol and triterpene biosynthesis have been cloned and characterized from various plants. For example, β-amyrin synthase and cycloartenol synthase from *P. ginseng* were characterized by Kushiro et al. [7]. Five OSCs from *Kalanchoe daigremontiana* were reported by Wang et al. [8]. Kim et al. [9] characterized dammaranediol synthase from *Centella asiatica*. Several other OSCs were also reported, such as β-amyrin synthase from *Aster sedifolius* and *Arabidopsis thaliana* [10] and OSC from *Aster tataricus* [11] and from tomato [12]. Dhar et al. [13] isolated three branch-point OSCs from *Withania somnifera*. Sandeep et al. [4] reported four OSCs from the medicinal tree banaba, and Guo et al. [5] reported cycloartenol synthase from *Paris polyphylla*. All these OSCs have been cloned and functionally characterized in the mutant yeast strain GIL77, which has lanosterol synthase deficiency. 

*Hoodia gordonii* (Masson) Sweet ex Decne., of the family Apocynaceae, is a leafless, spiny succulent plant with medicinal properties. The indigenous people of the Kalahari Desert in South Africa and Namibia used the plant as an appetite suppressant and to treat indigestion and small infections. Due to its appetite-depressant properties, the genus has become a popular dietary supplement for weight loss. The active ingredient P57AS3, a steroidal glycoside responsible for the putative appetite-suppressant effect, was patented by CSIR South Africa in 2002 (U.S. Patent No. 6376657). Chemical fingerprints and LC-MS/UV detection were developed to identify P57AS3. Furthermore, more than 70% of the components in the extracts from the whole plant were identified as steroid glycosides that consisted of a triterpenoid or steroidal skeleton [14]. Despite this progress at the chemical level, very little or no progress has been initiated at the molecular level for producing such specialized metabolites. To our knowledge, none of the genes have yet been cloned and characterized for triterpene or sterol biosynthesis from *H. gordonii*. The current study was initiated by searching public RNA-seq data repositories to find out the expressed gene members of the *Hoodia* OSCs gene family. The two newly identified oxidosqualene cyclase genes from *H. gordonii* were cloned and functionally characterized using recombinant technology. 

## 2. Results

### 2.1. Isolation and Identification of cDNA Clones for Triterpene Synthases

Published transcriptomic and metabolomic data for *H. gordonii* on the Medicinal Plant Genome Resources (MPGR) server (http://mpgr.uga.edu/ (accessed on 25 June 2019)) have significantly expanded resources for molecular biology research for this species. To mine the *H. gordonii* transcriptome data, we downloaded the RNA sequences from MPGR. A total of 41,793 transcripts were annotated, and the unique sequences in the dataset were assigned to GO terms (Figure 1a). Of particular interest, “other metabolic processes” was the largest assignment within the “biological process” category (44.85% of all terms assigned), suggesting a high level of metabolic activity for this plant species. A KEGG analysis of the selected metabolic pathways and biosynthesis of secondary metabolites indicated that 99 transcripts were associated with terpenoid biosynthesis (Figure 1b). We then performed BLAST searches against the transcriptomic dataset using a cycloartenol synthase (CAS) sequence from *Arabidopsis thaliana* as the query sequence. Nine homologs were identified, and two predicted transcripts that showed considerable similarity to established plant OSCs were prioritized. Attempts to obtain analogous full-length ORFs through the rapid amplification of cDNA ends (RACE) were thus initiated using high-quality total RNAs prepared from stem tissue. The corresponding full-length cDNA clones were then isolated and designated as *HgOSC1* and *HgOSC2*, respectively. The ORF of *HgOSC1* was 2.289 kb and that of *HgOSC2* was 2.277 kb. BLASTn analysis for the *HgOSC1* sequence exhibited 76% similarity to lupeol synthase from *Olea europaea* and *HgOSC2* showed 79% similarity to *Centella asiatica* cycloartenol synthase. 

### 2.2. In Silico Characterization and Phylogenetic Analysis

The ORFs coding for *HgOSC1* and *HgOSC2* corresponded to proteins of 87 (762 amino acids) and 86 (758 amino acids) kDa, respectively. The amino acid sequence of HgOSC1 showed 79% similarity to lupeol synthase from *Vitis vinifera*, while 83% sequence identity was obtained for HgOSC2 with cycloartenol synthase from *Sesamum indicum* and *Panax quinquefolius*. The NCBI conserved domain search indicated that these Hoodia OSCs possess the conserved superfamily domains SQCY_1 (cd02892) and “ISOPREN_C2_like”. Proteins having these domains belong to class II terpene cyclases including 2,3-oxidosqualene cyclases (Appendix A). 

The two OSC proteins were aligned with previously published plant lupeol and cycloartenol synthases and analyzed in MegAlign Pro (DNA Star v15). Both the cloned OSCs possess the amino acid motif DCTAE that is associated with substrate binding and the four characteristic motifs of the OSC superfamily. The prenyl transferase and squalene oxidase repeat Motif A are present in both OSCs. The conserved Motif B (MLCYCR) is present only in HgOSC1 and lupeol synthases, and absent in HgOSC2 and cycloartenol synthases from other plants. The analysis also revealed the presence of conserved catalytic aspartic acid (Motif C) in both HgOSC1 and HgOSC2. The conserved Motif C is responsible for converting squalene into a carbocation, which is necessary for initiating ring cyclization by protonating the first C=C bond. Lastly, the highly conserved Motif D, the QW motif, is present in both lupeol and cycloartenol synthases (Appendix A).

Phylogenetic analyses using the neighbor-joining tree method showed that the HgOSC1 clusters together with the previously characterized lupeol synthases, while the HgOSC2 most closely clusters with the previously described cycloartenol synthases (Figure 2). 

### 2.3. Functional Characterization of OSCs in Yeast

For biochemical characterization of the putative OSCs from *H. gordonii*, the full-length ORFs of these cDNA clones were heterologously expressed in *Saccharomyces cerevisiae*. The yeast mutant *GIL77* strain (Wang et al., 2011 [12]) was used for this expression as it lacks lanosterol synthase and accumulates a high concentration of 2,3-oxidosqualene which can serve as the substrate for OSC-catalyzed cyclization. However, external ergosterol must be added to the medium to grow the mutant strain. The *HgOSC* ORF was subcloned into the yeast expression vector *pYES2*, and the transgenic construct was transformed into the mutant yeast *GIL77* strain. An empty vector was also transformed in the mutant yeast strain to serve as a negative control. The induced yeast cultures were pelleted and saponified with 6% methanolic KOH, and the hexane-extractable fractions were analyzed via GC-MS to identify different triterpenoids. All the transformants were found to contain squalene and ergosterol. In addition, the extracts from yeast expressing *HgOSC1* cDNA contained one more compound, identified as lupeol (Figure 3 and Appendix A). The extracts were further separated by thin-layer chromatography (TLC). The TLC chromatogram showed the presence of lupeol in the hexane extract (Appendix A). The lupeol structure was confirmed by comparing the mass spectra with the commercial standard in ^1^H and ^13^C NMR spectral analyses (Appendix A). On the other hand, our initial attempt at the characterization of *HgOSC2* in the mutant yeast was unsuccessful. The expression of *HgOSC2* in the mutant yeast did not generate detectable new products in the yeast extracts compared to the control strain harboring the empty vector. However, the gene recoding of *HgOSC2* to optimize codon usage (Appendix A) for the expression in yeast in the mutant strain led to activity. The extracts from the strain harboring this recoded *HgOSC2* cDNA exhibited many compounds that were not present in the empty vector control, with one being cycloartenol (Figure 4 and Appendix A). Among the other identified compounds were squalene, ergosterol, and cycloartenol acetate. Most of the additional compounds in the yeast extract expressing *HgOSC2* are yet to be identified. Nevertheless, this result suggests that HgOSC2 may be a multifunctional triterpene synthase.

### 2.4. Yeast Spotting Assay

The growth of yeast cells harboring the *pYES2-HgOSC2* construct was examined on a selective medium (Figure 5). Growth assessment in the form of yeast spotting assays was conducted after five days. On the culture medium supplied with ergosterol, yeast harboring either *pYES2-HgOSC2* or empty vector exhibited similar growth profiles along different ranges of dilutions (1, 0.1, 0.01, and 0.001). However, in the absence of ergosterol, the cells containing *pYES2-HgOSC2* grew significantly compared to those with vector pYES2. However, the growth of the yeast transformants with *pYES2-HgOSC2* slowed down as indicated by the reduced number of spots at higher dilutions (Figure 5).

### 2.5. Functional Expression of HgOSC1 and HgOSC2 in Nicotiana benthamiana

To validate the results obtained by the expression of the *Hoodia* oxidosqualene cyclases in yeast, constructs for plant transformation were prepared to express full-length cDNAs coding for *HgOSC1* and *HgOSC2* under the control of the constitutive cauliflower mosaic virus (CaMV) 35S promoter and were mobilized into *Agrobacterium tumefaciens*. Transient expression was performed in *N. benthamiana* plants using *A. tumefaciens* harboring the plasmids. Triterpene products from transiently expressed *HgOSC1* and *HgOSC2* were extracted 6 days after *A. tumefaciens* infiltration and analyzed via GC-MS. As shown in Figure 6, consistent with yeast expression results, the transient expression of *HgOSC1* in *N. benthamiana* produced a compound not detected in the extracts of the leaves infiltrated with the empty vector control. The triterpene product was identified as lupeol via GC-MS by comparing its retention time and MS characteristics with an authentic standard. However, cycloartenol was not detected in the sample extracted from the leaves transiently expressing *HgOSC2*. 

## 3. Discussion

*Hoodia gordonii* is endemic to the Kalahari Desert in South Africa and Namibia. The indigenous people used the Hoodia plants as appetite suppressants and medicinal herb to treat various ailments. *H. gordonii* is one of the top dietary supplements marketed for weight loss. Most of the studies on this species mainly focused on the active chemical molecule P57, a steroidal glycoside, and other secondary metabolites, such as pregnane glycosides. However, no studies are available on the molecular characterization of the genes involved in triterpene and steroid metabolic pathways. To our knowledge, this is the first investigation on the cloning and molecular characterization of genes responsible for triterpene or sterol biosynthesis from *H. gordonii*. According to our initial study, lupeol and cycloartenol accumulated in the young tissues of *H. gordonii.* However, a detailed understanding of biosynthetic pathways for lupeol and cycloartenol is required to improve their production. The common precursor for the sterols and triterpenes is 2,3-oxidosqualene (OS). The cyclization of OS into sterols and triterpenes is catalyzed by oxidosqualene cyclases (OSCs). Numerous 2,3-oxidosqualene cyclase genes encoding oxidosqualene cyclases responsible for the production of triterpenes and sterols have been isolated from a wide range of plant species [4,7,12,13]. We have also successfully cloned and characterized two genes (*HgOSC1* and *HgOSC2*) of the OSC superfamily that belong to dammarenyl and protosteryl cation groups, respectively, from *H. gordonii*. Both the cloned OSCs were found to be closely related to *A. thaliana* OSCs and possessed the amino acid motif DCTAE, which is associated with substrate binding and the four characteristics of QW motifs that are features of the OSC superfamily [15]. The phylogenetic analyses using the neighbor-joining tree method showed that both the HgOSCs were clustered with their respective previously characterized lupeol and cycloartenol synthases, thus confirming their homology. 

The functional characterization of HgOSC1 in the lanosterol-deficient yeast strain *GIL77* yielded lupeol as the major cyclization product. The GC-MS analysis of the HgOSC1 compound, along with the reference standard, confirmed that the expressed product was lupeol. NMR spectral analyses were carried out to further confirm the structure of lupeol by comparing it with ^1^H and ^13^C NMR spectral data of the reference standard. Hence, it was concluded that HgOSC1 encodes the lupeol synthase in *H. gordonii*. The expression pattern of monofunctional *Hoodia* lupeol synthase is similar to what has been previously reported for other plants such as *Eleutherococcus trifoliatus* [16], *Withania somnifera* [13], and *Cleome arabica* [17]. However, when HgOSC2 was expressed in the *GIL77* yeast strain, the cyclization of the 2,3-oxidosqualene resulted in multiple products, with one of the cyclization products being identified as cycloartenol, although it was not a major product (Figure 4). After the cyclization events, how 2,3-oxidosqualene is folded into the enzyme active site also determines the diversity of end products. The two conformations, chair–boat–chair (CBC) and chair–chair–chair (CCC) generate protosteryl and dammarenyl cations, respectively. The protosteryl cation leads to the formation of cycloartenol, which is the precursor of phytosterols and steroid hormones. In the case of *Hoodia*, cycloartenol synthase was found to be a multifunctional enzyme. In most plants, cycloartenol synthases are monofunctional enzymes, e.g., in pea [18], *Fritillaria thunbergii* [19], *Withania somnifera* [13] and *Paris polyphylla* [5]. On the other hand, some plants like Kalanchoe produce cycloartenol along with some other side products when OSCs are expressed in yeast [8]. Similarly, in the present study, the heterologous expression of *Hoodia* OSC (HgOSC2) in yeast resulted in the production of cycloartenol along with cycloartenol acetate, squalene, and ergosterol, as well as other side compounds that are yet to be identified (Figure 4). The multifunctionality of such an enzyme may contribute to the wide range and high total steroid glycoside levels (>70% of plant extracts) in *Hoodia* plants [14]. To further confirm the cycloartenol synthase expression in yeast, the growth of yeast harboring *pYES2-HgOSC2* was examined via the spotting assay on a minimal selective medium in the presence or absence of ergosterol. In the presence of ergosterol, yeast containing *pYES2* or *pYES2-HgOSC2* showed a similar growth profile along a range of dilutions. However, in the absence of ergosterol, only the yeast cells expressing HgOSC2 could grow as compared to the empty vector. The reduced number of spots evidenced the slow growth of the yeast cells. This reinforces the previous studies showing that cycloartenol could support yeast growth to some extent in the absence of lanosterol and supports further that cycloartenol could be metabolized to C-4 desmethyl sterols [2]. The function of HgOSC1 in plants was confirmed by transient expression in *N. benthamiana*. However, cycloartenol was not detected in the sample extracted from the leaves transiently expressing HgOSC2. Since, cycloartenol was not a major product, as shown in the results of the expression in mutant yeast cells (Figure 4), it can be challenging to identify cycloartenol in the HgOSC2-infiltrated leaves under the current experimental conditions.

## 4. Materials and Methods

### 4.1. Plant Materials

Authenticated *H. gordonii* plant materials were collected from the Maynard W. Quimby Medicinal Plant Garden at the University of Mississippi. The plants were grown in a greenhouse under controlled temperature and humidity. The young and mature shoot tissues were collected during April, flash-frozen and stored at −80 °C until further use.

### 4.2. Isolation of RNA

RNA isolation was performed using the Qiagen RNA isolation kit (Qiagen, Valencia, CA, USA) as per the manufacturer’s directions. The isolated RNAs were treated with RNase-free DNase to remove possible contaminant DNA using the RNeasy Mini Kit (Qiagen, Valencia, CA, USA). RNA quantity and quality were determined using the NanoDrop ND-1000 Spectrophotometer (Thermo Fisher Scientific, Waltham, MA, USA). RNA sample integrity was also assessed via agarose gel electrophoresis. 

### 4.3. Functional Annotation of the Transcriptome Dataset and Pathway Analyses

The transcriptome data were downloaded from the Medicinal Plant Metabolomics Resource website (http://mpgr.uga.edu/ (accessed on 25 June 2019)). Functional annotation was performed using the custom pipeline on the Linux computing cluster at the Chinese Academy of Medical Sciences and Peking Union Medical College. The unique putative transcripts were aligned against the NCBI non-redundant nucleotide using BLASTN [20] with an E-value cutoff of 10^−5^. BlastX was performed to search protein databases including NCBI Non-redundant (Nr) and Swiss-Prot (https://www.uniprot.org/ (accessed on 26 June 2019)). TAIR Gene Ontology (TAIR-GO, http://www.arabidopsis.org (accessed on 26 June 2019)) analyses were performed using a custom script to execute gene categories. Each transcript annotated to TAIR was classified based on TAIR GO terms. The percentage of each functional categorization was computed by the TAIR GO functional categorization method, which represents the percentage of the number of genes annotated to terms in the GO slim category and the total number of transcripts. Metabolic pathways were identified using KEGG (http://www.genome.jp/kegg/pathway.html (accessed on 28 June 2019)) pathway mapper using PERL script. The annotated transcripts were mapped to the specific pathways based on KEGG Ontology (KO) groups and the KEGG pathway maps.

### 4.4. Isolation of cDNA Clones for Terpene Synthases

To obtain full-length cDNA clones, the rapid amplification of both 5′ and 3′ cDNA ends (RACE) was carried out using the BD SMART™ RACE cDNA Amplification Kit (Clontech, Mountain View, CA, USA) according to the manufacturer’s instructions. Different primer pairs were used for the RACE PCR to amplify the two selected genes (Appendix A). The amplified products with single bands were extracted from the gel, and the purified cDNA was cloned into the TOPO-TA cloning vector. The clones were sent for Sanger’s sequencing (GENEWIZ, Burlington, MA, USA) to confirm the amplified product. The full-length cDNAs were then amplified with primer pairs complementary to the 5′ and 3′ ends of the open reading frames (ORFs) identified in RACE experiments using Pfu thermostable DNA polymerase (Stratagene, La Jolla, CA, USA) and first-strand cDNA generated from *H*. *gordonii* plants. ProtoScript M-MULV first strand cDNA synthesis kit (New England Biolabs, Ipswich, MA, USA) was used to prepare the cDNAs. The *HgOSC1* and *HgOSC2* sequences reported in this article have been deposited in the GenBank database (accession nos. *HgOSC1*, OR133751; *HgOSC2*, OR133752). 

### 4.5. Phylogenetic Analysis

HgOSC1 and HgOSC2 protein sequences were compared with functionally characterized oxidosqualene cyclases from other plant gene sequences retrieved from GenBank via phylogenetic comparison. Following protein alignments using ClustalW, the neighbor-joining tree was constructed with five hundred bootstrap iterations in MEGAX software version 10 [21]. 

### 4.6. Expression of OSCs in Saccharomyces cerevisiae

The cDNA encoded in the ORFs of *HgOSC1* and *HgOSC2* was ligated to *pYES2* expression vector via XbaI and KpnI sites for *HgOSC1* and via KpnI and XhoI sites for *HgOSC2* to generate *pYES2-HgOSC1* and *pYES2-HgOSC2*, respectively. Lanosterol-deficient strain *GIL77* (gal2 hem3-6 erg7 ura3-167), provided by Dr. Reinhard Jetter (The University of British Columbia), was transformed either with an empty vector or with *pYES2-HgOSC1* and *pYES2-HgOSC2* constructs. The ORF of *HgOSC2* was synthesized via GenScript (Piscataway, NJ, USA) for optimized codon usage for yeast and used for the expression experiments due to lack of detectable activity of the original clone. The lithium-acetate method [22] was used to transform these plasmids into the mutant yeast strain *GIL77*. The transformed yeast cells were selected for the uracil prototype and were grown at 30 °C for three days in the presence of exogenous ergosterol and hemin. The functional characterization of these yeast strains was carried out according to the methods previously described (Wang et al., 2011 [12]). Briefly, the expression of OSCs was induced by galactose (2%) added to the minimal medium along with ergosterol and incubated at 30 °C for 16 hrs. The induced cells were then transferred to a resting medium containing phosphate buffer (pH 7.0) with glucose and hemin. The cells were incubated for one day at 30 °C and then pelleted down with 3500 rpm centrifugation for 5 min. The pelleted cells were frozen at −80 °C for later use in GC-MS analysis. 

### 4.7. GC-MS Analysis

To isolate and analyze the metabolites, the frozen yeast cells (approximately 100 mg) were saponified in 6% potassium hydroxide methanolic solution at 80 °C for 1 h. They were then extracted twice in hexane and dried in a Speedvac. The sample was resuspended in 500 µL methylene chloride prior to GC/Q-ToF analysis. Samples were analyzed on an Agilent 7890B GC (Santa Clara, CA, USA) equipped with an Agilent PAL RSI85 auto-sampler. The mass spectral detector was an Agilent 7250 High-Resolution Accurate Mass Q-TOF MS operated in the full spectral acquisition mode. The Agilent J&W HP-5MS UI capillary columns (30 m × 0.25 mm i.d. × 0.25 µm film thickness) were used for the analysis. The carrier gas was helium with a 1 mL/min flow rate. The GC oven temperature was held at 50 °C for 2 min, programmed at 30 °C/min to 200 °C, then ramped up at 3 °C/min to 300 °C and held for 10 min. One microliter of each sample was injected with a split-less injection mode. The inlet temperature was fixed at 280 °C.

The Q-ToF mass spectrometer was equipped with an electron ionization source, operated with an electron energy of 70 eV and an emission current of 5.0 µA. The source, quadrupole, and transfer line temperatures were set to 280 °C, 150 °C, and 280 °C, respectively. Data were acquired at a rate of 5 Hz from 45 to 550 *m*/*z* with a 5 minute solvent delay. Automated ToF mass calibration was performed after every injection using a keyword command in the sequence table. Data were acquired using Agilent MassHunter Acquisition Software (version B.10.0). Data analysis was performed using Agilent MassHunter Qualitative (version B.10.0) and Quantitative (version B.10.0) analysis software. The identification of compounds began with a comparison to the spectral databases (Wiley and NIST) using a probability-based matching algorithm. The identification was confirmed based on standard references purchased from commercial sources. 

### 4.8. TLC and NMR Analysis of HgOSC1 Products

The transformed yeast *GIL77* strain with *pYES2-HgOSC1* construct and the *pYES2* plasmid without any gene insert that served as control were grown on a minimal medium for three days. A few colonies of each were grown in 500 mL of minimal medium lacking uracil and supplemented with ergosterol and hemin. The yeast cultures were grown with shaking at 210 rpm at 30 °C for two days. The cells were then pelleted down via centrifugation (3500 rpm) for 5 min. The pellet was then resuspended in 500 mL induction medium containing 2% galactose supplemented with ergosterol and hemin. The cells were grown in the induction medium for sixteen hours. After induction, the cells were transferred to a resting medium. An equal volume of 6% methanolic KOH was added to the culture to extract the triterpenes. The mixture was then incubated at 80 °C with shaking (200 rpm) for one hour. The unsaponifiable fractions were then extracted with an equal volume of hexane, followed by drying in a rotary evaporator without heating.

To determine the products of the HgOSC1 clone, the hexane extract was then analyzed via thin-layer chromatography (TLC) to detect the presence of lupeol. The TLC was performed on the GF254 silica gel 60 plate (EMD Millipore) with chloroform/methanol (10:0.1, *v*/*v*) as the solvent system for development. The reference standard of lupeol (Sigma-Aldrich, St. Louis, MO, USA) was spotted on the plate beside the extract for the purpose of identification. The isolate zones were observed using UV light (254 nm) and then visualized by spraying with 1% vanillin-H_2_SO_4_ reagent followed by heating. To further verify the presence and identity of lupeol produced by HgOSC1, nuclear magnetic resonance (NMR) analysis was conducted for the hexane extract. ^1^H and ^13^C NMR spectra at 25 °C were obtained with a OneNMR probe on an Agilent DD2-500 NMR spectrometer (Santa Clara, CA, USA). The ^1^H and ^13^C NMR spectra of the hexane extract and the lupeol standard’s spectra were stacked for comparison. 

### 4.9. Yeast Spotting Assay

Isolated yeast colonies containing either *pYES2* empty vector or *pYES2-HgOSC2* were grown overnight at 30 °C in yeast dropout medium (SGal) supplemented with ergosterol and hemin. Yeast cells were pelleted, washed, and resuspended in sterile water to reach an OD_600_ equal to 1. Serial dilutions were made as 1, 0.1, 0.01, and 0.001 in sterile water, and 5 µL of each was spotted on SGal agar media with or without exogenous ergosterol and hemin. The plates were incubated at 30 °C for five days.

### 4.10. Transient Expression of Hoodia OSCs in Nicotiana benthamiana

The agrobacterium T-DNA binary vectors for the expression of the *Hoodia* OSCs were constructed according to the methods previously described [23]. Briefly, the ORFs of *HgOSC1* and *HgOSC2* were amplified using PfuUltra HS DNA polymerase (Agilent) and subcloned into the shuttle vector between the 2x35S promoter and the octopine synthase terminator (Tosc). The expression cassettes were then subcloned into pLH7000 [24] using the flanking SfiI sites to generate the binary expression vectors *pLH-OSC1* and *pLH-OSC2*, respectively. The constructs were then mobilized into *Agrobacterium tumefaciens* strain EHA105. Four-week-old *N. benthamiana* leaves were co-infiltrated with the strain expressing either *pLH-OSC1* or *pLH-OSC2* along with the strain expressing the suppressor of silencing, p19, in a ratio of 3:1 at an OD_600_ of 0.6 in the infiltration buffer (10 mM 2-(N-morpholino) ethanesulfone, 10 mM MgCl2, pH 5.7, and 100 μM acetosyringone) by injection into the lower leaf epidermis using a 1 mL syringe. Plants were maintained in a Conviron growth chamber under a 16 h/8 h, light/dark cycle at 25 °C, 150 µmol m^−2^s^−1^ light intensity. Agroinfiltrated *N. benthamiana* leaves were collected 6 days post-infiltration, ground to a fine powder under liquid N_2_ in a mortar and pestle, and stored at −80 °C until analysis. 

In total, 100 mg of powdered frozen *N. benthamiana* leaf sample was extracted with 600 µL of methanol/chloroform (2:1) twice via centrifugation at 1000× *g* for 5 min. The resultant supernatants were combined and dried using a centrifugal vacuum evaporator. The dried sample was then saponified in 300 μL of methanol/10% potassium hydroxide (4:1, *v*/*v*) for 1 h at room temperature with shaking at 150 rpm. Then, 600 μL water was added to the saponified sample and extracted three times with 300 μL hexane. The hexane fractions were pooled, washed three times with 300 μL water, and then evaporated to dryness. The dried hexane extract was resuspended in 100 μL of 100% methanol for GC-MS analysis. The experiments were independently repeated three times.

## 5. Conclusions

In conclusion, two different OSCs have been isolated from *Hoodia gordonii* and characterized via heterologous expression in yeast. The two different triterpenoids, lupeol and cycloartenol, were identified as the products of the enzymes. This is the first report on the identification and biochemical characterization of lupeol and cycloartenol synthase from *Hoodia*. This study will provide an opportunity to elucidate the metabolic pathways of sterol and triterpenoid production in this particular plant species. 

## Figures and Tables

**Figure 1 plants-13-00231-f001:**
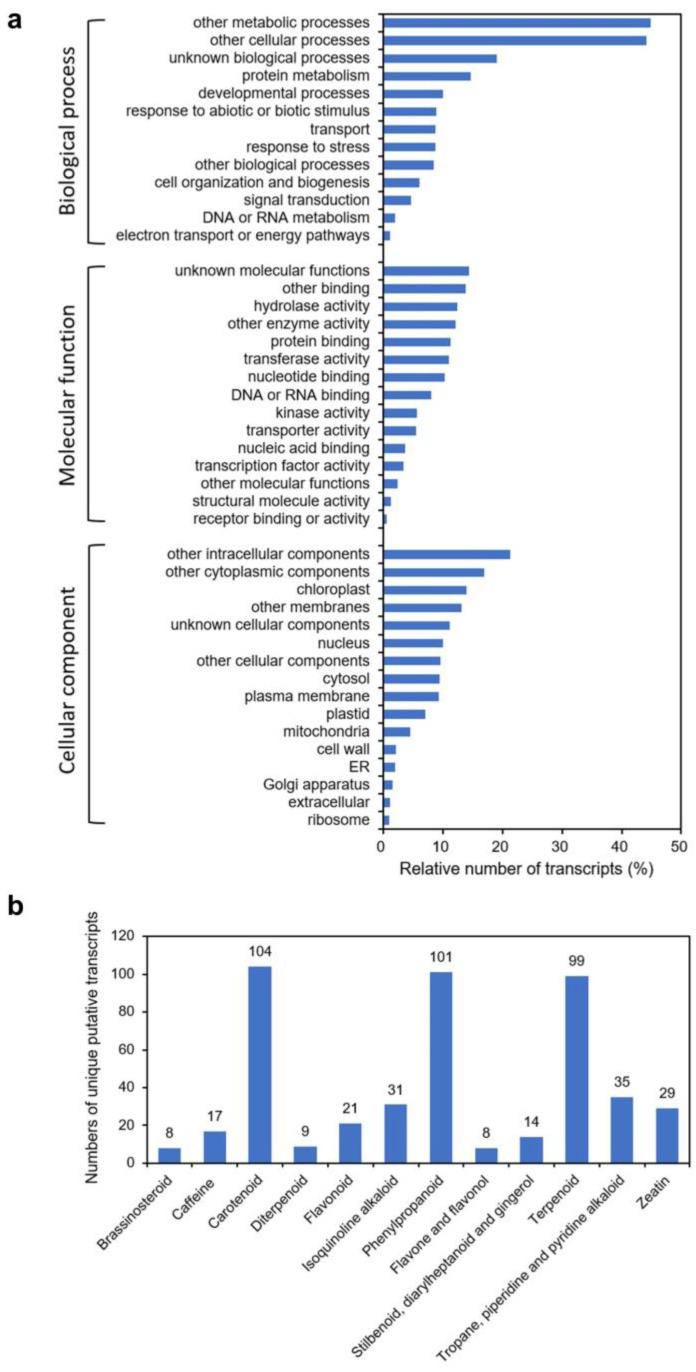
Distribution of unique putative transcripts annotated in the *Hoodia gordonii* transcriptome dataset. (**a**) Functional annotation of transcripts from *H. gordonii* based on GO categories. The percentage within each GO category (biological process, molecular function, and cellular component) is indicated on the x-axis. (**b**) KEGG classification of biosynthetic pathways. The numbers of the unique transcripts related to selected secondary metabolic pathways are shown.

**Figure 2 plants-13-00231-f002:**
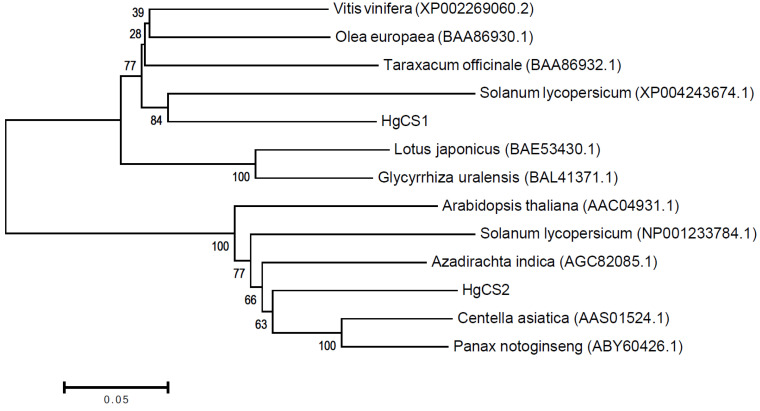
Phylogenetic relationships of selected functionally characterized oxidosqualene cyclases from plants. Amino acid sequences were aligned in ClustalW, and the phylogenetic tree was constructed using the neighbor-joining method with 500 bootstrap iterations in MEGAX. HgOSC1 showed a close relationship with lupeol synthases, whereas HgOSC2 formed a clade with other previously characterized cycloartenol synthases.

**Figure 3 plants-13-00231-f003:**
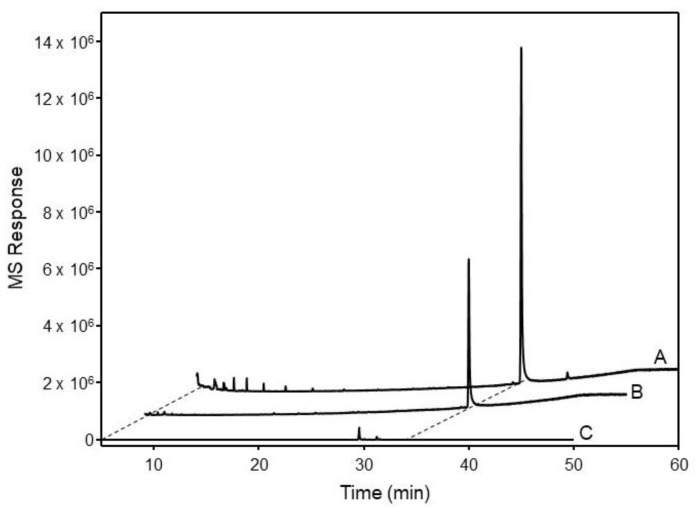
Identification of lupeol in transgenic yeast. GC/MS analyses showed peaks of the cell extracts from mutant yeast expressing *HgOSC1* (B) and the empty vector pYES2 control (C) vs. lupeol as a reference standard (A). Results are depicted as extracted ion chromatograms (EICs) of the parent mass *m*/*z* 426.

**Figure 4 plants-13-00231-f004:**
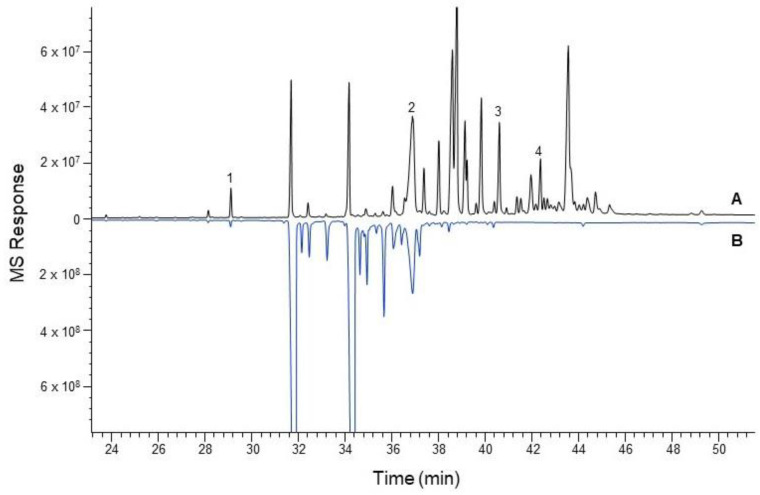
Total ion chromatogram (TIC) of GC/MS analysis of yeast cell extracts. TICs of extracts from the yeast strain transformed with *pYES2*-*HgOSC2* (A) and vector pYES2 as a control (B). Compounds identified were: 1, squalene; 2, ergosterol; 3, cycloartenol; and 4, cycloartenol acetate. The compound identification was based on the spectra search with NIST and Wiley libraries, and further confirmed with reference standards.

**Figure 5 plants-13-00231-f005:**
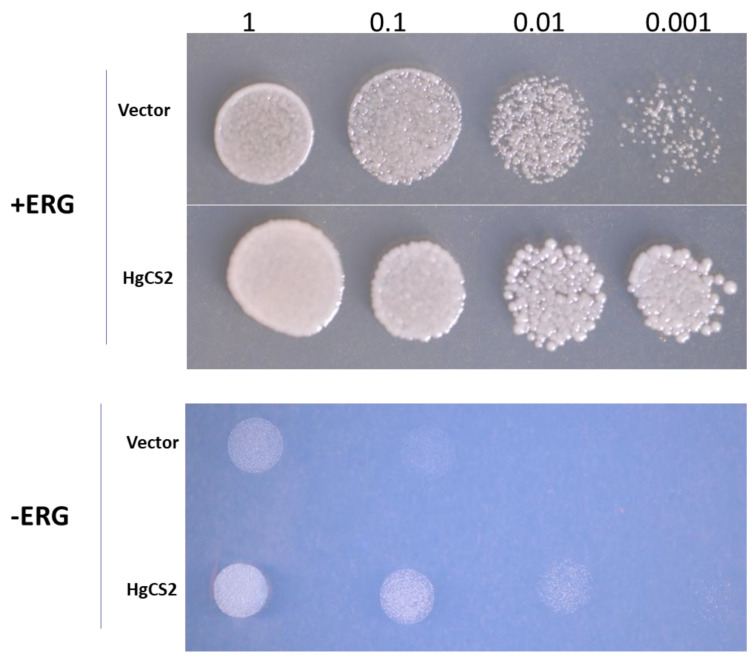
Yeast spotting assays with lanosterol deficient strain *GIL77*. Growth of *GIL77* yeast strains transformed with vector *pYES2* or construct *pYES2-HgOSC2* was tested on SGal medium supplemented with (+ERG) or without (−ERG) exogenous ergosterol. After 5 days of incubation, the plates supplemented with ergosterol showed fully grown colonies in both yeast strains, whereas Sgal plates without ergosterol allowed only a limited growth of yeast expressing *HgOSC2*, indicating that *HgOSC2* can partially overcome ergosterol deficiency.

**Figure 6 plants-13-00231-f006:**
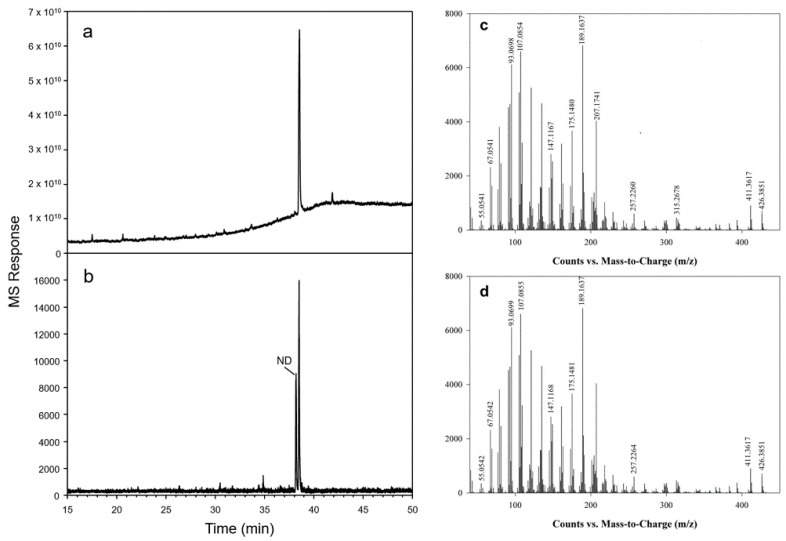
Expression of *Hoodia* oxidosqualene cyclase (HgOSC1) in *Nicotiana benthamiana*. *HgOSC1* transiently co-expressed with the suppressor of the gene silencing P19 in *N. benthamiana* leaves by agroinfiltration. Leaf metabolites were extracted (see Materials and Methods) and analyzed via GC-MS (**b**), depicted as EIC, vs. lupeol reference standard (**a**). Leaves expressing P19 with an empty vector only served as an experimental control. Mass spectra of products corresponding to the peaks shown in a (**c**) and b (**d**). The experiments were repeated three times independently with identical results. ND, not determined.

## Data Availability

Data are contained within the article and Appendix A.

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
