# Peer review of "Identification and Functional Characterization of Oxidosqualene Cyclases from Medicinal Plant *Hoodia gordonii"

_plants, 2024, doi:10.3390/plants13020231_

Round 1

Reviewer 1 Report

Comments and Suggestions for Authors

In this manuscript entitled “Identification and Functional Characterization of Oxidosqualene Cyclases from Medicinal Plant Hoodia gordonii”, the study aimed to find the enzymes involved in the biosynthesis of triterpenes and phytosterols from H. gordonii at the molecular level. These studies revealed that distinct OSCs exist for triterpene formation in H. gordonii and it provided opportunities for metabolic engineering of the specific precursors in the production of phytosterols. Generally, the research is relatively in-depth and can be accepted for publication. However, there are still some issues to be solved.

1. More information on collected plant samples is required, such as season, temperature, etc.

2. In this manuscript, two different OSCs have been isolated from Hoodia gordonii and characterized by heterologous expression in yeast. Should the electropherogram of Oxidosqualene cyclases (OSCs)be provided?

3. Line 450, “ian yeast” should be “in yeast”.

Author Response

In this manuscript entitled “Identification and Functional Characterization of Oxidosqualene Cyclases from Medicinal Plant Hoodia gordonii”, the study aimed to find the enzymes involved in the biosynthesis of triterpenes and phytosterols from H. gordonii at the molecular level. These studies revealed that distinct OSCs exist for triterpene formation in H. gordonii and it provided opportunities for metabolic engineering of the specific precursors in the production of phytosterols. Generally, the research is relatively in-depth and can be accepted for publication. However, there are still some issues to be solved.

Author's response: Thank you so  much for the detailed and rigorous review. Below is the point wise response to your comments/ edits.

  1. More information on collected plant samples is required, such as season, temperature, etc.

Author’s response: Thank you for the comment. The following sentence is added:

The plants were grown in green house under controlled temperature and humidity. The young and mature shoot tissues were collected during April-May, were flash-frozen and stored at -80°C until further use.

  1. In this manuscript, two different OSCs have been isolated from Hoodia gordonii and characterized by heterologous expression in yeast. Should the electropherogram of Oxidosqualene cyclases (OSCs)be provided?

Author’s response: Thank you for the comment. The HgOSC1 and HgOSC2 sequences reported in this article have been deposited in the GenBank database (accession nos. HgOSC1, OR133751; HgOSC2, OR133752)

  1. Line 450, “ian yeast” should be “in yeast”.

Author’s response: Thank you for the comment. Line 450 is corrected

Reviewer 2 Report

Comments and Suggestions for Authors

In this paper, the predicted transcripts potentially encoding oxidosqualene cyclases were recognized first by searching publicly available H. gordonii RNA-seq data sets. Two OSC-like sequences were selected for functional analysis. A monofunctional OSC, designated HgOSC1 which encodes lupeol synthase, and HgOSC2, a multifunctional cycloartenol synthase forming cycloartenol and other products, were observed through recombinant enzyme studies.

1. How did you determine the HgOSC1 the content by GC-MS? Was the internal standard compound used?

2. In Figure 4, besides the peaks 1-4, there are others peaks that were not identified? Why? What are they?

3. The results from GC-MS and NMR should be discussed.

4. The formats of references in text body are not correct, it should be authors et al. (year), not authors et al. (authors, year).

5. Line 167, for 1 h; Line 239, 100 mg; line 240, 1000 × g (×, not letter x; italic for g), 5 min.

Author Response

In this paper, the predicted transcripts potentially encoding oxidosqualene cyclases were recognized first by searching publicly available H. gordonii RNA-seq data sets. Two OSC-like sequences were selected for functional analysis. A monofunctional OSC, designated HgOSC1 which encodes lupeol synthase, and HgOSC2, a multifunctional cycloartenol synthase forming cycloartenol and other products, were observed through recombinant enzyme studies.

Author's response: Thank you so much for the detailed and rigorous review. Below is the point wise response to your comments/ edits.

  1. How did you determine the HgOSC1 the content by GC-MS? Was the internal standard compound used?

Author’s response: Thank you for the query. Yes, the Lupeol internal standard was used to confirm the compound synthesized by HgOSC1.

  1. In Figure 4, besides the peaks 1-4, there are others peaks that were not identified? Why? What are they?

Author’s response: Thank you for the query. The standards for peaks 1-4 were available and hence were identified. The other peaks are intermediate compounds of sterol synthesis for which the standards were not found.

  1. The results from GC-MS and NMR should be discussed.

Author’s response: Thank you for the comment. The following paragraph is added to the discussion

The GC-MS analysis of HgOSC1 compound along with the internal standard confirmed that the expressed products is Lupeol. To further confirm the structure of lupeol NMR spectral analyses was carried out. The structure was confirmed by comparing the mass spectra with the commercial standard in 1H and 13C NMR.

  1. The formats of references in text body are not correct, it should be authors et al. (year), not authors et al. (authors, year).

Author’s response: Thank you for the comment. The correction is made in the text body and highlighted.

  1. Line 167, for 1 h; Line 239, 100 mg; line 240, 1000 × g (×, not letter x; italic for g), 5 min.

Author’s response: Thank you for the comment. All the suggested edits were incorporated in the text.

Reviewer 3 Report

Comments and Suggestions for Authors

The researchers examined the roles of two oxidosqualene cyclases (OSCs) cloned from Hoodia plant, namely HgOSC1 and HgOSC2. The experiments were well designed and executed, and the manuscripts were expertly prepared.

Comments on the Quality of English Language

Two comments:

Line 172-173, you may want to remove “…two 15m x 0.25 mm x 0.25 µm Agilent HP-5MS Ultra Insert column.” The GC columns used has been included in the following sentence.

Line 450, “…expression in yeast.” 

Author Response

The researchers examined the roles of two oxidosqualene cyclases (OSCs) cloned from Hoodia plant, namely HgOSC1 and HgOSC2. The experiments were well designed and executed, and the manuscripts were expertly prepared.

Author's response: Thank you so much for the detailed and rigorous review. Below is the point wise response to your comments/ edits.

Comments on the Quality of English Language

Two comments:

Line 172-173, you may want to remove “…two 15m x 0.25 mm x 0.25 µm Agilent HP-5MS Ultra Insert column.” The GC columns used has been included in the following sentence.

Author’s response: Thank you for the comment. It has been deleted.

Line 450, “…expression in yeast.” 

Author’s response: Thank you for the comment. It is corrected in the text.